# E(n) Equivariant Normalizing Flows

**Victor Garcia Satorras**[1*], **Emiel Hoogeboom**[1*], **Fabian B. Fuchs**[2],
**Ingmar Posner**[2], **Max Welling**[1]
UvA-Bosch Delta Lab, University of Amsterdam[1],
Department of Engineering Science, University of Oxford[2]
v.garciasatorras@uva.nl,e.hoogeboom@uva.nl,fabian@robots.ox.ac.uk

## Abstract

This paper introduces a generative model equivariant to Euclidean symmetries: E($n$) Equivariant Normalizing Flows (E-NFs). To construct E-NFs, we take the discriminative E($n$) graph neural networks and integrate them as a differential equation to obtain an invertible equivariant function: a continuous-time normalizing flow. We demonstrate that E-NFs considerably outperform baselines and existing methods from the literature on particle systems such as DW4 and LJ13, and on molecules from QM9 in terms of log-likelihood. To the best of our knowledge, this is the first flow that jointly generates molecule features and positions in 3D.

## 1 Introduction

Leveraging the structure of the data has long been a core design principle for building neural networks. Convolutional layers for images famously do so by being translation equivariant and therefore incorporating the symmetries of a pixel grid. Analogously, for discriminatory machine learning tasks on 3D coordinate data, taking into account the symmetries of data has significantly improved performance (Thomas et al., 2018; Anderson et al., 2019; Finzi et al., 2020; Fuchs et al., 2020; Klicpera et al., 2020). One might say that equivariance has been proven an effective tool to build inductive bias into the network about the concept of 3D coordinates. However, for generative tasks, e.g., sampling new molecular structures, the development of efficient yet powerful rotation equivariant approaches—while having made great progress—is still in its infancy.

A recent method called E($n$) Equivariant Graph Neural Networks (EGNNs) (Satorras et al., 2021) is both computationally cheap and effective in regression and classification tasks for molecular data, while being equivariant to Euclidean symmetries. However, this model is only able to discriminate features on nodes, and cannot generate new molecular structures.

In this paper, we introduce E($n$) Equivariant Normalizing Flows (E-NFs): A generative model for E($n$) Equivariant data such as molecules in 3D. To construct E-NFs we parametrize a continuous-time flow, where the first-order derivative is modelled by an EGNN. We adapt EGNNs so that they are stable when utilized as a derivative. In addition, we use recent advances in the dequantization literature to lift the discrete features of nodes to a continuous space. We show that our proposed flow model significantly outperforms its non-equivariant variants and previous equivariant generative methods (Köhler et al., 2020). Additionally, we apply our method to molecule generation and we show that our method is able to generate realistic molecules when trained on the QM9 dataset.

---

[*]Equal contribution.

35th Conference on Neural Information Processing Systems (NeurIPS 2021).

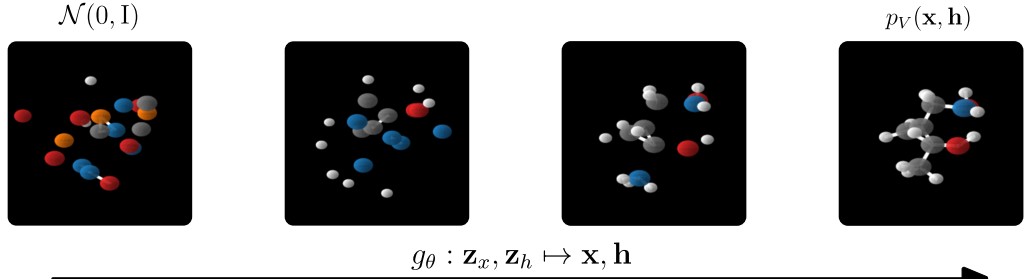

$\mathcal{N}(0, \mathrm{I})$

$p_V(\mathbf{x}, \mathbf{h})$

$g_\theta : \mathbf{z}_x, \mathbf{z}_h \mapsto \mathbf{x}, \mathbf{h}$

Figure 1: Overview of our method in the sampling direction. An equivariant invertible function $g_\theta$ has learned to map samples from a Gaussian distribution to molecules in 3D, described by $\mathbf{x}, \mathbf{h}$.

## 2 Background

**Normalizing Flows**  A learnable invertible transformation $\mathbf{x} = g_\theta(\mathbf{z})$ and a simple base distribution $p_Z$ (such as a normal distribution) yield a complex distribution $p_X$. Let $f_\theta = g_\theta^{-1}$, then the likelihood of a datapoint $\mathbf{x}$ under $p_X$ can be exactly computed using the change of variables formula:

$$p_X(\mathbf{x}) = p_Z(\mathbf{z}) \left| \det J_f(\mathbf{x}) \right|, \text{ where } \mathbf{z} = f(\mathbf{x}), \tag{1}$$

where $J_f$ is the Jacobian of $f_\theta$. A particular type of normalizing flows are *continuous-time* normalizing flows (Chen et al., 2017, 2018b; Grathwohl et al., 2018). These flows use a conceptual time direction to specify an invertible transformation as a differential equation. The first order differential is predicted by a neural network $\phi$, referred to as the dynamics. The continuous-time change of variables formula is given by:

$$\log p_X(\mathbf{x}) = \log p_Z(\mathbf{z}) + \int_0^1 \mathrm{Tr}\, J_\phi(\mathbf{x}(t)) \mathrm{d}t, \text{ where } \mathbf{z} = \mathbf{x} + \int_0^1 \phi(\mathbf{x}(t)) \mathrm{d}t, \tag{2}$$

where $\mathbf{x}(0) = \mathbf{x}$ and $\mathbf{x}(1) = \mathbf{z}$. In practice, Chen et al. (2018b); Grathwohl et al. (2018) estimate the trace using Hutchinson's trace estimator, and the integrals can be straightforwardly computed using the `torchdiffeq` package written by Chen et al. (2018b). It is often desired to regularize the dynamics for faster training and more stable solutions (Finlay et al., 2020). Continuous-time normalizing flows are desirable because the constraints that need to be enforced on $\phi$ are relatively mild: $\phi$ only needs to be high order differentiable and Lipschitz continuous, with a possibly large Lipschitz constant.

**Equivariance in Normalizing Flows**  Note that in this context, the desirable property for distributions is often invariance whereas for transformations it is equivariance. Concretely, when a function $g_\theta$ is equivariant and a base distribution $p_Z$ is invariant, then the distribution $p_X$ given by $\mathbf{x} = g_\theta(\mathbf{z})$ where $\mathbf{z} \sim p_Z$ is also invariant (Köhler et al., 2020). As a result, one can design expressive invariant distributions using an invariant base distribution $p_Z$ and an equivariant function $g_\theta$. Furthermore, when $g_\theta$ is restricted to be bijective and $f_\theta = g_\theta^{-1}$, then equivariance of $g_\theta$ implies equivariance of $f_\theta$. In addition, the likelihood $p_X$ can be directly computed using the change of variables formula.

### 2.1 Equivariance

Equivariance of a function $f$ under a group $G$ is defined as $T_g(f(\mathbf{x})) = f(S_g(\mathbf{x}))$ for all $g \in G$, where $S_g, T_g$ are transformations related to the group element $g$, where in our case $S_g = T_g$ will be the same. In other words, $f$ being equivariant means that first applying the transformation $S_g$ on $x$ and then $f$ yields the same result as first applying $f$ on $x$ and then transforming using $T_g$.

In this paper we focus on symmetries of the $n$-dimensional Euclidean group, referred to as E($n$). An important property of this group is that its transformations preserve Euclidean distances. The transformations can be described as rotations, reflections and translations. Specifically, given a set of points $\mathbf{x} = (\mathbf{x}_1, \dots, \mathbf{x}_M) \in \mathbb{R}^{M \times n}$ embedded in an $n$-dimensional space, an orthogonal matrix $\mathbf{R} \in \mathbb{R}^{n \times n}$ and a translation vector $\mathbf{t} \in \mathbb{R}^n$, we can say $f$ is rotation (and reflection) equivariant if, for $\mathbf{z} = f(\mathbf{x})$ we have $\mathbf{R}\mathbf{z} = f(\mathbf{R}\mathbf{x})$, where $\mathbf{R}\mathbf{x}$ is the shorthand[2] for $(\mathbf{R}\mathbf{x}_1, \dots, \mathbf{R}\mathbf{x}_M)$. Similarly $f$

---

[2]To be precise, in matrix multiplication notation $(\mathbf{R}\mathbf{x}_1, \dots, \mathbf{R}\mathbf{x}_M) = \mathbf{x}\mathbf{R}^T$, for simplicity we use the notation $\mathbf{R}\mathbf{x} = (\mathbf{R}\mathbf{x}_1, \dots, \mathbf{R}\mathbf{x}_M)$ and see $\mathbf{R}$ as an operation instead of a direct multiplication on the entire $\mathbf{x}$.

is translation equivariant if $\mathbf{z} + \mathbf{t} = f(\mathbf{x} + \mathbf{t})$, where $\mathbf{x} + \mathbf{t}$ is the shorthand for $(\mathbf{x}_1 + \mathbf{t}, \ldots, \mathbf{x}_M + \mathbf{t})$. In this work we consider data defined on vertices of a graph $\mathcal{V} = (\mathbf{x}, \mathbf{h})$, which in addition to the position coordinates $\mathbf{x}$, also includes node features $\mathbf{h} \in \mathbb{R}^{M \times \text{nf}}$ (for example temperature or atom type). Features $\mathbf{h}$ have the property that they are *invariant* to E($n$) transformations, while they do affect $\mathbf{x}$. In other words, rotations and translations of $\mathbf{x}$ do not modify $\mathbf{h}$. In summary, given a function $f : \mathbf{x}, \mathbf{h} \mapsto \mathbf{z}_x, \mathbf{z}_h$ we require equivariance of $f$ with respect to the Euclidean group E($n$) so that for all orthogonal matrices $\mathbf{R}$ and translations $\mathbf{t}$:

$$\mathbf{R}\mathbf{z}_x + \mathbf{t}, \mathbf{z}_h = f(\mathbf{R}\mathbf{x} + \mathbf{t}, \mathbf{h}) \tag{3}$$

**E(n) Equivariant Graph Neural Networks (EGNN)** (Satorras et al., 2021) consider a graph $\mathcal{G} = (\mathcal{V}, \mathcal{E})$ with nodes $v_i \in \mathcal{V}$ and edges $e_{ij}$. Each node $v_i$ is associated with a position vector $\mathbf{x}_i$ and node features $\mathbf{h}_i$ as the ones defined in previous paragraphs. Then, an E($n$) Equivariant Graph Convolutional Layer (EGCL) takes as input the set of node embeddings $\mathbf{h}^l = \{\mathbf{h}_0^l, \ldots, \mathbf{h}_{M-1}^l\}$, coordiante embeddings $\mathbf{x}^l = \{\mathbf{x}_0^l, \ldots, \mathbf{x}_{M-1}^l\}$ at layer $l$ and edge information $\mathcal{E} = (e_{ij})$ and outputs a transformation on $\mathbf{h}^{l+1}$ and $\mathbf{x}^{l+1}$. Concisely: $\mathbf{h}^{l+1}, \mathbf{x}^{l+1} = \text{EGCL}[\mathbf{h}^l, \mathbf{x}^l, \mathcal{E}]$. This layer satisfies the equivariant constraint defined in Equation 3. The equations that define this layer are the following:

$$\mathbf{m}_{ij} = \phi_e\left(\mathbf{h}_i^l, \mathbf{h}_j^l, \left\|\mathbf{x}_i^l - \mathbf{x}_j^l\right\|^2\right) \quad \text{and} \quad \mathbf{m}_i = \sum_{j \neq i} e_{ij}\mathbf{m}_{ij}, \tag{4}$$

$$\mathbf{x}_i^{l+1} = \mathbf{x}_i^l + \sum_{j \neq i}\left(\mathbf{x}_i^l - \mathbf{x}_j^l\right)\phi_x\left(\mathbf{m}_{ij}\right) \quad \text{and} \quad \mathbf{h}_i^{l+1} = \phi_h\left(\mathbf{h}_i^l, \mathbf{m}_i\right). \tag{5}$$

A neural network $\mathbf{h}^L, \mathbf{x}^L = \text{EGNN}[\mathbf{h}^0, \mathbf{x}^0]$ composed of $L$ layers will define the dynamics of our ODE flow. In our experiments we are not provided with an adjacency matrix, but only node information. Therefore, we use the edge inference module $e_{ij} = \phi_{\text{inf}}(\mathbf{m}_{ij})$ introduced in the EGNN paper that outputs a soft estimation of the edges where $\phi_{\text{inf}} : \mathbb{R}^{\text{nf}} \to [0,1]^1$ resembles a linear layer followed by a sigmoid function. This behaves as an attention mechanism over neighbor nodes.

## 3 Related Work

Group equivariant neural networks (Cohen and Welling, 2016, 2017; Dieleman et al., 2016) have demonstrated their effectiveness in a wide variety of tasks. A growing body of literature is finding neural networks that are equivariant to transformations in Euclidean space (Thomas et al., 2018; Fuchs et al., 2020; Horie et al., 2020; Finzi et al., 2020; Hutchinson et al., 2020; Satorras et al., 2021). These methods have proven their efficacy in discriminative tasks and modelling dynamical sytems. On the other hand, graph neural networks (Bruna et al., 2013; Kipf and Welling, 2016) can be seen as networks equivariant to permutations. Often, the methods that study Euclidean equivariance operate on point clouds embedded in Euclidean space, and so they also incorporate permutation equivariance.

Normalizing Flows (Rippel and Adams, 2013; Rezende and Mohamed, 2015; Dinh et al., 2015) are an attractive class of generative models since they admit exact likelihood computation and can be designed for fast inference and sampling. Notably Chen et al. (2018a,b); Grathwohl et al. (2018) introduced continuous-time normalizing flows, a flow that is parametrized via a first-order derivative over time. This class of flows is useful because of the mild constraints on the parametrization function compared to other flow approaches (Dinh et al., 2017; Kingma and Dhariwal, 2018).

There are several specialized methods for molecule generation: Gebauer et al. (2019) generate 3D molecules iteratively via an autoregressive approach, but discretize positions and use additional focus tokens which makes them incomparable in log-likelihood. Gómez-Bombarelli et al. (2016); You et al. (2018); Liao et al. (2019) generate discrete graph structures instead of a coordinate space and Xu et al. (2021) generate molecule positions only. Some flows in literature Noé et al. (2018); Li et al. (2019); Köhler et al. (2020) model positional data, but not in combination with discrete properties.

Recently various forms of normalizing flows for equivariance have been proposed: Köhler et al. (2019, 2020) propose flows for positional data with Euclidean symmetries, Rezende et al. (2019) introduce a Hamiltonian flow which can be designed to be equivariant using an invariant Hamiltonian function. Liu et al. (2019); Biloš and Günnemann (2020) model distributions over graphs, with flows equivariant to permutations. Boyda et al. (2020) introduce equivariant flows for SU($n$) symmetries. For adversarial networks, Dey et al. (2021) introduced a generative network equivariant to 90 degree

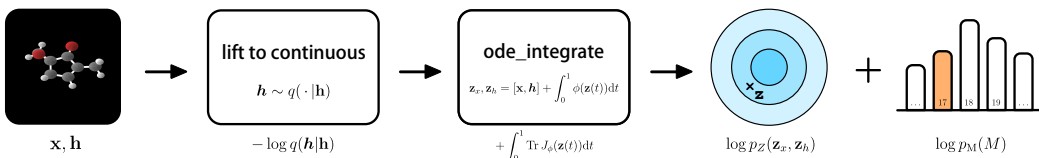

Figure 2: Overview of the training procedure: The discrete $\mathbf{h}$ is lifted to continuous $\boldsymbol{h}$. Then the variables $\mathbf{x}, \boldsymbol{h}$ are transformed by an ODE to $\mathbf{z}_x, \mathbf{x}_h$. To get a lower bound on $\log p_V(\mathbf{x}, \mathbf{h})$ we sum the variational term $-\log q(\boldsymbol{h}|\mathbf{h})$, the volume term from the ODE $\int_0^1 \mathrm{Tr}\, J_\phi(\mathbf{z}(t))\mathrm{d}t$, the log-likelihood of the latent representation on a Gaussian $\log p_Z(\mathbf{z}_x, \mathbf{x}_h)$, and the log-likelihood of the molecule size $\log p_\mathrm{M}(M)$. To train the model, the sum of these terms is maximized.

image rotations and reflections. Our work differs from these approaches in that we design a general-purpose normalizing flow for sets of nodes that contain both positional *and* invariant features while remaining equivariant to E($n$) transformations. This combination of positional and non-positional information results in a more expressive message passing scheme that defines the dynamics of our ODE. A by-product of our general approach is that it also allows us to jointly model features and structure of molecules in 3D space without any additional domain knowledge.

## 4 Method: E(n) Equivariant Normalizing Flows

In this section we propose our E($n$) Equivariant Normalizing Flows, a probabilistic model for data with Euclidean symmetry. The generative process is defined by a simple invariant distribution $p_Z(\mathbf{z}_x, \mathbf{z}_h)$ (such as a Gaussian) over a latent representation of positions $\mathbf{z}_x \in \mathbb{R}^{M \times n}$ and a latent representation of invariant features $\mathbf{z}_h \in \mathbb{R}^{M \times \mathrm{nf}}$ from which an invertible transformation $g_\theta(\mathbf{z}_x, \mathbf{z}_h) = (\mathbf{x}, \mathbf{h})$ generates $\mathcal{V} = (\mathbf{x}, \mathbf{h})$. These nodes consist of node features $\mathbf{h} \in \mathbb{R}^{M \times \mathrm{nf}}$ and position coordinates $\mathbf{x} \in \mathbb{R}^{M \times n}$ embedded in a $n$-dimensional space. Together, $p_Z$ and $g_\theta$ define a distribution $p_V$ over node positions and invariant features.

Since we will restrict $g_\theta$ to be invertible, $f_\theta = g_\theta^{-1}$ exists and allows us to map from data space to latent space. This makes it possible to directly optimize the implied likelihood $p_V$ of a datapoint $\mathcal{V}$ utilizing the change of variables formula:

$$p_V(\mathcal{V}) = p_V(\mathbf{x}, \mathbf{h}) = p_Z(f_\theta(\mathbf{x}, \mathbf{h}))|\det J_f| = p_Z(\mathbf{z}_x, \mathbf{z}_h)|\det J_f|, \tag{6}$$

where $J_f = \frac{\mathrm{d}(\mathbf{z}_x, \mathbf{z}_h)}{\mathrm{d}\mathcal{V}}$ is the Jacobian, where all tensors are vectorized for the Jacobian computation. The goal is to design the model $p_V$ such that translations, rotations and reflections of $\mathbf{x}$ do not change $p_V(\mathbf{x}, \mathbf{h})$, meaning that $p_V$ is E($n$) invariant with respect to $\mathbf{x}$. Additionally, we want $p_V$ to also be invariant to permutations of $(\mathbf{x}, \mathbf{h})$. A rich family of distributions can be learned by choosing $p_Z$ to be invariant, and $f_\theta$ to be equivariant.

**The Normalizing Flow** As a consequence, multiple constraints need to be enforced on $f_\theta$. From a probabilistic modelling perspective 1) we require $f_\theta$ to be invertible, and from a symmetry perspective 2) we require $f_\theta$ to be equivariant. One could utilize the EGNN (Satorras et al., 2021) which is E($n$) equivariant, and adapt the transformation so that it is also invertible. The difficulty is that when both these constraints are enforced naïvely, the space of learnable functions might be small. For this reason we opt for a method that only requires mild constraints to achieve invertibility: neural ordinary differential equations. These methods require functions to be Lipschitz (which most neural networks are in practice, since they operate on a restricted domain) and to be continuously differentiable (which most smooth activation functions are).

To this extent, we define $f_\theta$ to be the differential equation, integrated over a conceptual time variable using a differential predicted by the EGNN $\phi$. We redefine $\mathbf{x}, \mathbf{h}$ as functions depending on time, where $\mathbf{x}(t = 0) = \mathbf{x}$ and $\mathbf{h}(t = 0) = \mathbf{h}$ in the data space. Then $\mathbf{x}(1) = \mathbf{z}_x, \mathbf{h}(1) = \mathbf{z}_h$ are the latent representations. This admits a formulation of the flow $f$ as the solution to an ODE defined as:

$$\mathbf{z}_x, \mathbf{z}_h = f(\mathbf{x}, \mathbf{h}) = [\mathbf{x}(0), \mathbf{h}(0)] + \int_0^1 \phi(\mathbf{x}(t), \mathbf{h}(t))\mathrm{d}t. \tag{7}$$

The solution to this equation can be straightforwardly obtained by using the `torchdiffeq` package, which also supports backpropagation. The Jacobian term under the ODE formulation is $\log|\det J_f| = \int_0^1 \operatorname{Tr} J_\phi(\mathbf{x}(t), \mathbf{h}(t))\mathrm{d}t$ as explained in Section 2, Equation 2. The Trace of $J_\phi$ has been approximated with the Hutchinson's trace estimator.

**The Dynamics**    The dynamics function $\phi$ in Equation 7 models the first derivatives of $\mathbf{x}$ and $\mathbf{h}$ with respect to time, over which we integrate. Specifically: $\frac{\mathrm{d}}{\mathrm{d}t}\mathbf{x}(t), \frac{\mathrm{d}}{\mathrm{d}t}\mathbf{h}(t) = \phi(\mathbf{x}(t), \mathbf{h}(t))$. This derivative is modelled by the EGNN of $L$ layers introduced in Section 2.1:

$$\frac{\mathrm{d}}{\mathrm{d}t}\mathbf{x}(t), \frac{\mathrm{d}}{\mathrm{d}t}\mathbf{h}(t) = \mathbf{x}^L(t) - \mathbf{x}(t), \mathbf{h}^L(t) \quad \text{where} \quad \mathbf{x}^L(t), \mathbf{h}^L(t) = \text{EGNN}[\mathbf{x}(t), \mathbf{h}(t)]. \quad (8)$$

Notice that we directly consider the output $\mathbf{h}^L$ from the last layer $L$ of the EGNN as the differential $\frac{\mathrm{d}}{\mathrm{d}t}\mathbf{h}(t)$ of the node features, since the representation is invariant. In contrast, the differential of the node coordinates is computed as the difference between the EGNN output and intput $\frac{\mathrm{d}}{\mathrm{d}t}\mathbf{x}(t) = \mathbf{x}^L - \mathbf{x}(t)$. This choice is consistent with the nature of velocity-type equivariance: although $\frac{\mathrm{d}}{\mathrm{d}t}\mathbf{x}(t)$ rotates exactly like $\mathbf{x}$, it is unaffected by translations as desired.

The original EGNN from (Satorras et al., 2021) is unstable when utilized in an ODE because the coordinate update from Equation 5 would easily explode. Instead, we propose an extension in Equation 9 that normalizes the relative difference of two coordinates by their norm plus a constant $C$ to ensure differentiability. In practice we set $C = 1$ and found this to give stable results.

$$\mathbf{x}_i^{l+1} = \mathbf{x}_i^l + \sum_{j \neq i} \frac{(\mathbf{x}_i^l - \mathbf{x}_j^l)}{\|\mathbf{x}_i^l - \mathbf{x}_j^l\| + C}\phi_x(\mathbf{m}_{ij}) \quad (9)$$

**Translation Invariance**    Recall that we want the distribution $p_V(\mathcal{V})$ to be translation invariant with respect to the overall location and orientation of positional coordinates $\mathbf{x}$. For simplicity, let's assume only a distribution $p_X(\mathbf{x})$ over positions and an invertible function $\mathbf{z} = f(\mathbf{x})$. Translation invariance is defined as $p_X(\mathbf{x} + \mathbf{t}) = p_X(\mathbf{x})$ for all $\mathbf{t}$: a constant function. However, this cannot be a distribution since it cannot integrate to one. Instead, we have to restrict $p_X$ to a subspace.

To construct a translation invariant $p_X$, we can restrict the data, flow $f_\theta$ and prior $p_Z$ to a translation invariant linear subspace, for instance by centering the nodes so that their center of gravity is zero. Then the positions $\mathbf{x} \in \mathbb{R}^{M \times n}$ lie on the $(M-1) \times n$-dimensional linear subspace defined by $\sum_{i=1}^M \mathbf{x}_i = \mathbf{0}$. However, from a modelling perspective it is easier to represent node positions as $M$ sets of coordinates that are $n$-dimensional in the ambient space. In short, we desire the distribution to be defined on the subspace, but with the representation of the nodes in the ambient space.

To limit the flow to the subspace, in practice only the mean of the output of the dynamics network $\phi$ is removed, following (Köhler et al., 2020). Expanding their analysis, we derive that the Jacobian determinant in the ambient space is indeed equal to the Jacobian determinant in the subspace under this condition. Intuitively, the transformation $f_\theta$ does not change orthogonal to the subspace, and as a result there is no volume change in that direction. For this reason the determinant can safely be computed in the ambient space, which conveniently allows the use of existing libraries without modification. Additionally, we can find the proper normalization constant for the base distribution.

For a more formal argument, let $P$ be a $\mathbb{R}^{(M-1)\cdot n \times M \cdot n}$ matrix that projects points to the $(M-1) \cdot n$ dimensional subspace with orthonormal rows. Consider a collection of points $\mathbf{x} \in \mathbb{R}^{M \times n}$ where $\sum_{i=1}^M \mathbf{x}_i = \mathbf{0}$ and $\mathbf{z} = f_\theta(\mathbf{x})$. Define $\tilde{\mathbf{x}} = P\mathbf{x}$, $\tilde{\mathbf{z}} = P\mathbf{z}$, where $\mathbf{x}, \mathbf{z}$ are considered to be vectorized and $\tilde{\cdot}$ signifies that the variable is defined in the coordinates of the subspace. The Jacobian in the subspace $\tilde{J}_f$ is:

$$\tilde{J}_f = \frac{\mathrm{d}\tilde{\mathbf{z}}}{\mathrm{d}\tilde{\mathbf{x}}} = \frac{\mathrm{d}\tilde{\mathbf{z}}}{\mathrm{d}\mathbf{z}}\frac{\mathrm{d}\mathbf{z}}{\mathrm{d}\mathbf{x}}\frac{\mathrm{d}\mathbf{x}}{\mathrm{d}\tilde{\mathbf{x}}} = PJ_fP^T. \quad (10)$$

To connect the determinant of $J_f$ to $\tilde{J}_f$, let $Q \in \mathbb{R}^{M \cdot n \times M \cdot n}$ be the orthogonal extension of $P$ using orthonormal vectors $\mathbf{q}_1, \ldots, \mathbf{q}_n$, so that $Q^T = \begin{bmatrix} P^T & \mathbf{q}_1 \ldots \mathbf{q}_n \end{bmatrix}$. Then $QJ_fQ^T = \begin{bmatrix} \tilde{J}_f & 0 \\ 0 & I_n \end{bmatrix}$, where $\tilde{J}_f = PJ_fP^T$ and $I_n$ an $n \times n$ identity matrix. From this we observe that $\det J_f = \det QJ_fQ^T = \det \tilde{J}_f$, which proves the claim. As a result of this argument, we are able to utilize existing methods to compute volume changes in the subspace without modification, as $\det \tilde{J}_f = \det J_f$, under the constraint that $f_\theta$ is an identity orthogonal to the subspace.

**The base distribution**    Next we need to define a base distribution $p_Z$. This base distribution can be divided in two parts: the positional part $p(\mathbf{z}_x)$ and the feature part $p(\mathbf{z}_h)$, which we will choose to be independent so that $p(\mathbf{z}_x, \mathbf{z}_h) = p(\mathbf{z}_x) \cdot p(\mathbf{z}_h)$. The feature part is straightforward because the features are already invariant with respect to $E(n)$ symmetries, and only need to be permutation invariant. A common choice is a standard Gaussian $p(\mathbf{z}_h) = \mathcal{N}(\mathbf{z}_h|0, \mathbf{I})$. For the positional part recall that $\mathbf{z}_x$ lies on an $(M-1)n$ subspace, and we need to specify the distribution over this space. Standard Gaussian distributions are reflection and rotation invariant since $||R\mathbf{z}_x||^2 = ||\mathbf{z}_x||^2$ for any rotation or reflection $R$. Further, observe that for our particular projection $\tilde{\mathbf{z}}_x = P\mathbf{z}_x$ it is true that $||\tilde{\mathbf{z}}_x||^2 = ||\mathbf{z}_x||^2$ since $\mathbf{z}_x$ lies in the subspace. More formally this can be seen using the orthogonal extension $Q$ of $P$ as defined earlier and observing that: $||\mathbf{z}_x||^2 = ||Q\mathbf{z}_x||^2 = \left|\left| \begin{bmatrix} \tilde{\mathbf{z}}_x \\ \mathbf{0} \end{bmatrix} \right|\right|^2 = ||\tilde{\mathbf{z}}_x||^2$.

Therefore, a valid choice for a rotation invariant base distribution on the subspace is given by:

$$p(\tilde{\mathbf{z}}_x) = \mathcal{N}(\tilde{\mathbf{z}}_x|0, \mathbf{I}) = \frac{1}{(2\pi)^{(M-1)n/2}} \exp\left( -\frac{1}{2}||\mathbf{z}_x||^2 \right), \tag{11}$$

which can be directly computed in the ambient space using $||\mathbf{z}_x||^2$, with the important property that the normalization constant uses the dimensionality of the subspace: $(M-1)n$, so that the distribution is properly normalized.

**Modelling discrete properties**    Normalizing flows model continuous distributions. However, the node features $\mathbf{h}$ may contain both ordinal (e.g. charge) and categorical (e.g. atom type) features. To train a normalizing flow on these, the values need to be lifted to a continuous space. Let $\mathbf{h} = (\mathbf{h}_{\mathrm{ord}}, \mathbf{h}_{\mathrm{cat}})$ be divided in ordinal and categorical features. For this we utilize variational dequantization (Ho et al., 2019) for the ordinal features and argmax flows (Hoogeboom et al., 2021) for the categorical features. For the ordinal representation $\mathbf{h}_{\mathrm{ord}}$, interval noise $\boldsymbol{u} \sim q_{\mathrm{ord}}(\cdot|\mathbf{h}_{\mathrm{ord}})$ is used to lift to the continuous representation $\boldsymbol{h}_{\mathrm{ord}} = \mathbf{h}_{\mathrm{ord}} + \boldsymbol{u}$. Similarly, $\mathbf{h}_{\mathrm{cat}}$ is lifted using a distribution $\boldsymbol{h}_{\mathrm{cat}} \sim q_{\mathrm{cat}}(\cdot|\mathbf{h}_{\mathrm{cat}})$ where $q_{\mathrm{cat}}$ is the probabilistic inverse to an argmax function. Both $q_{\mathrm{ord}}$ and $q_{\mathrm{cat}}$ are parametrized using normal distributions where the mean and standard deviation are learned using an EGNN conditioned on the discrete representation. This formulation allows training on the continuous representation $\boldsymbol{h} = (\boldsymbol{h}_{\mathrm{ord}}, \boldsymbol{h}_{\mathrm{cat}})$ as it lowerbounds an implied log-likelihood of the discrete representation $\mathbf{h}$ using variational inference:

$$\log p_{\mathrm{H}}(\mathbf{h}) \geq \mathbb{E}_{\boldsymbol{h} \sim q_{\mathrm{ord,cat}}(\cdot|\mathbf{h})} \Big[ \log p_H(\boldsymbol{h}) - \log q_{\mathrm{ord,cat}}(\boldsymbol{h}|\mathbf{h}) \Big] \tag{12}$$

To sample the discrete $\mathbf{h} \sim p_{\mathrm{H}}$, first sample the continuous $\boldsymbol{h} \sim p_H$ via a flow and then compute $\mathbf{h} = (\mathrm{round}(\boldsymbol{h}_{\mathrm{ord}}), \mathrm{argmax}(\boldsymbol{h}_{\mathrm{cat}}))$ to obtain the discrete version. In short, instead of training directly on the discrete properties $\mathbf{h}$, the properties are lifted to the continuous variable $\boldsymbol{h}$. The lifting method depends on whether a feature is categorical or ordinal. On this lifted continuous variable $\boldsymbol{h}$ the flow learns $p_H$, which is guaranteed to be a lowerbound via Equation 12 on the discrete $p_{\mathrm{H}}$. To avoid clutter, in the remainder of this paper no distinction is made between $\mathbf{h}$ and $\boldsymbol{h}$ as one can easily transition between them using $q_{\mathrm{ord}}, q_{\mathrm{cat}}$ and the $\mathrm{round}, \mathrm{argmax}$ functions.

Finally, the number of nodes $M$ may differ depending on the data. In this case we extend the model using a simple one dimensional categorical distribution $p_{\mathrm{M}}$ of $M$ categories. This distribution $p_{\mathrm{M}}$ is constructed by counting the number of molecules and dividing by the total. The likelihood of a set of nodes is then $p_V(\mathbf{x}, \mathbf{h}, M) = p_{V_M}(\mathbf{x}, \mathbf{h}|M)p_{\mathrm{M}}(M)$, where $p_{V_M}(\mathbf{x}, \mathbf{h}|M)$ is modelled by the flow as before and the same dynamics can be shared for different sizes as the EGNN adapts to the number of nodes. In notation we sometimes omit the $M$ conditioning for clarity. To generate a sample, we first sample $M \sim p_{\mathrm{M}}$, then $\mathbf{z}_x, \mathbf{z}_h \sim p_Z(\mathbf{z}_x, \mathbf{z}_h|M)$ and finally transform to the node features and positions via the flow.

## 5    Experiments

### 5.1    DW4 and LJ13

In this section we study two relatively simple sytems, DW-4 and LJ-13 presented in (Köhler et al., 2020) where E($n$) symmetries are present. These datasets have been synthetically generated by sampling from their respective energy functions using Markov Chain Monte Carlo (MCMC).

**DW4**: This system consists of only M=4 particles embedded in a 2-dimensional space which are governed by an energy function that defines a coupling effect between pairs of particles with multiple metastable states. More details are provided in Appendix A.1.

**LJ-13**: This is the second dataset used in (Köhler et al., 2020) which is given by the *Leonnard-Jones* potential. It is an approximation of inter-molecular pair potentials that models repulsive and attractive interactions. It captures essential physical principles and it has been widely studied to model solid, fluid and gas states. The dataset consists of M=13 particles embedded in a 3-dimensional state. More details are provided in Appendix A.1.

Both energy functions (DW4 and LJ13) are equivariant to translations, rotations and reflections which makes them ideal to analyze the benefits of equivariant methods when E($n$) symmetries are present on the data. We use the same MCMC generated dataset from (Köhler et al., 2020). For both datasets we use 1,000 validation samples, and 1,000 test samples. We sweep over different numbers of training samples $\{10^2, 10^3, 10^4, 10^5\}$ and $\{10, 10^2, 10^3, 10^4\}$ for DW4 and LJ13 respectively to analyze the performance in different data regimes.

**Implementation details:** We compare to the state-of-the art E($n$) equivariant flows "Simple Dynamics" and "Kernel Dynamics" presented in (Köhler et al., 2020). We also compare to non-equivariant variants of our method, Graph Normalizing Flow (GNF), GNF with attention (GNF-att) and GNF with attention and data augmentation (GNF-att-aug), i.e. augmenting the data with rotations. Our E-NF method and its non-equivariant variants (GNF, GNF-att, GNF-att-aug) consist of 3 layers each, 32 features per layer, and SiLU activation functions. All reported numbers have been averaged over 3 runs. Further implementation details are provided in the Appendix A.1.

Table 1: Negative Log Likelihood comparison on the test partition over different methods on DW4 and LJ13 datasets for different amount of training samples averaged over 3 runs.

| | **DW4** | | | | **LJ13** | | | |
|---|---|---|---|---|---|---|---|---|
| # Samples | $10^2$ | $10^3$ | $10^4$ | $10^5$ | 10 | $10^2$ | $10^3$ | $10^4$ |
| GNF | 11.93 | 11.31 | 10.38 | 7.95 | 43.56 | 42.84 | 37.17 | 36.49 |
| GNF-att | 11.65 | 11.13 | 9.34 | 7.83 | 43.32 | 36.22 | 33.84 | 32.65 |
| GNF-att-aug | 8.81 | 8.31 | 7.90 | 7.61 | 41.09 | 31.50 | 30.74 | 30.93 |
| Simple dynamics | 9.58 | 9.51 | 9.53 | 9.47 | 33.67 | 33.10 | 32.79 | 32.99 |
| Kernel dynamics | 8.74 | 8.67 | 8.42 | 8.26 | 35.03 | 31.49 | 31.17 | 31.25 |
| E-NF | **8.31** | **8.15** | **7.69** | **7.48** | **33.12** | **30.99** | **30.56** | **30.41** |

**Results:** In Table 1 we report the cross-validated Negative Log Likelihood for the test partition. Our E-NF outperforms its non-equivariant variants (GNF, GNF-att and GNF-att-aug) and (Köhler et al., 2020) methods in all data regimes. It is interesting to see the significant increase in performance when including data augmentation (from GNF-att to GNF-att-aug) in the non-equivariant models.

## 5.2 QM9 Positional

We introduce QM9 Positional as a subset of QM9 that only considers positional information and does not encode node features. The aim of this experiment is to compare our method to those that only operate on positional data (Köhler et al., 2020) while providing a more challenging scenario than synthetically generated datasets. QM9 Positional consists only of molecules with 19 atoms/nodes, where each node only has a 3-dimensional positional vector associated. The likelihood of a molecule should be invariant to translations and rotations on a 3-dimensional space which makes equivariant models very suitable for this type of data. The dataset consists of 13,831 training samples, 2,501 for validation and 1,813 for test.

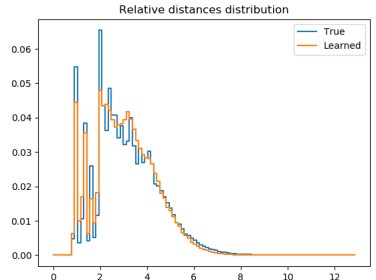

Figure 3: Normalized histogram of relative distances between atoms for QM9 Positional and E-NF generated samples.

In this experiment, in addition to reporting the estimated Negative Log Likelihood, we designed a a metric to get an additional insight into the quality of the generated samples. More specifically, we produce a histogram of relative distances between all pairs of nodes within each molecule and we

compute the Jensen–Shannon divergence (Lin, 1991) $JS_{div}(P_{gen}||P_{data})$ between the normalized histograms from the generated samples and from the training set. See Figure 3 for an example.

**Implementation details**: As in the previous experiment, we compare our E-NF to its non-equivariant variants GNF, GNF-att, GNF-att-aug and to the equivariant methods from (Köhler et al., 2020) Simple Dynamics and Kernel Dynamics. The dynamics of our E-NF, GNF, GNF-att and GNF-att-aug consist of 6 convolutional layers each, the number of features is set to 64 and all activation layers are SiLUs. The learning rate is set to $5 \cdot 10^{-4}$ for all experiments except for the E-NF and Simple dynamics which was set to $2 \cdot 10^{-4}$ for stability reasons. All experiments have been trained for 160 epochs. The JS divergence values have been averaged over the last 10 epochs for all models.

| # Metrics | NLL | JS(rel. dist) |
|---|---|---|
| **Simple dynamics** | 73.0 | .086 |
| **Kernel dynamics** | 38.6 | .029 |
| **GNF** | -00.9 | .011 |
| **GNF-att** | -26.6 | .007 |
| **GNF-att-aug** | -33.5 | .006 |
| **E-NF** (ours) | **-70.2** | .006 |

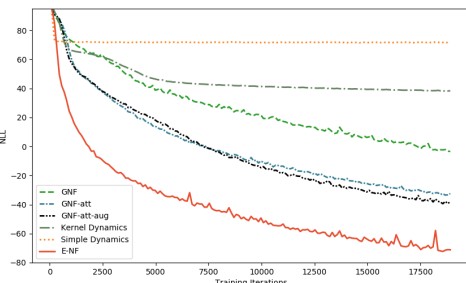

Figure 4: The table on the left presents the Negative Log Likelihood (NLL) $-\log p_V(\mathbf{x})$ for the QM9 Positional dataset on the test data. The figure on the right shows the training curves for all methods.

**Results:** In the table from Figure 4 we report the cross validated Negative Log Likelihood $-\log p_V(\mathbf{x})$ for the test data and the Jensen-Shannon divergence. Our E-NF outperforms all other algorithms in terms of NLL of the dataset. Additionally, the optimization curve with respect to the number of iterations converges much quicker for our E-NF than for the other methods as shown on the right in Figure 4. Regarding the JS divergence, the E-NF and GNF-att-aug achieve the best performance.

### 5.3 QM9 Molecules

QM9 (Ramakrishnan et al., 2014) is a molecular dataset standarized in machine learning as a chemical property prediction benchmark. It consists of small molecules (up to 29 atoms per molecule). Atoms contain positional coordinates embedded in a 3D space, a one-hot encoding vector that defines the type of molecule (H, C, N, O, F) and an integer value with the atom charge. Instead of predicting properties from molecules, we use the QM9 dataset to learn a generative distribution over molecules. We desire the likelihood of a molecule to be invariant to translations and rotations, therefore, our $E(n)$ equivariant normalizing flow is very suitable for this task. To summarize, we model a distribution over 3D positions $\mathbf{x}$, and atom properties $\mathbf{h}$. These atom properties consist of the atom type (categorical) and atom charge (ordinal).

We use the dataset partitions from (Anderson et al., 2019), 100K/18K/13K for training/validation/test respectively. To train the method, the nodes $(\mathbf{x}, \mathbf{h})$ are put into Equation 7 as $\mathbf{x}(0), \mathbf{h}(0)$ at time 0 and integrated to time 1. Using the continuous-time change of variables formula and the base distribution, the (negative) log-likelihood of a molecule is computed $-\log p_V(\mathbf{x}, \mathbf{h}, M)$. Since molecules differ in size, this term includes $-\log p_M(M)$ which models the number of atoms as a simple 1D categorical distribution and is part of the generative model as described in Section 4.

**Implementation details**: We compare our E-NF to the non-equivariant GNF-att and GNF-att-aug introduced in previous experiments. Notice in this experiment we do not compare to (Köhler et al., 2020) since this dataset contains invariant feature data in addition to positional data. Each model is composed of 6 layers, 128 features in the hidden layers and SilU activation functions. We use the same learning rates as in QM9 Positional. Note that the baselines can be seen as instances of permutation equivariant flows (Liu et al., 2019; Biloš and Günnemann, 2020) but where the GNN architectures have been chosen to be as similar as possible to the architecture in our E-NFs.

**Results (quantitative):** Results are reported in Table 2. As in previous experiments, our E-NF significantly outperforms the non-equivariant models GNF and GNF-aug. In terms of negative log-likelihood, the E-NF performs much better than its non-equivariant counterparts. One factor that increases this difference is the E-NFs ability to capture the very specific distributions of inter-atomic distances. Since the E-NF is able to better capture these sharp peaks in the distribution, the

Table 2: Neg. log-likelihood $-\log p_V(\mathbf{x}, \mathbf{h}, M)$, atom stability and mol stability for the QM9 dataset.

| # Metrics | NLL | Atom stability | Mol stable |
|---|---|---|---|
| **GNF-attention** | -28.2 | 72% | 0.3% |
| **GNF-attention-augmentation** | -29.3 | 75% | 0.5% |
| **E-NF** (ours) | **-59.7** | **85%** | **4.9%** |
| Data | - | 99% | 95.2% |

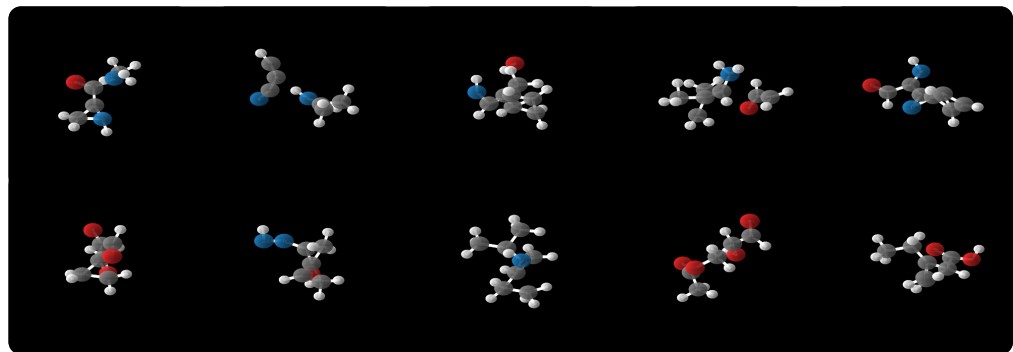

Figure 5: Sampled molecules by our E-NF. The top row contains random samples, the bottom row also contains samples but selected to be stable. Edges are drawn depending on inter-atomic distance.

negative log-likelihood becomes much lower. This effect is also seen when studying the number of stable atoms and molecules, which is very sensitive to the inter-atomic distances. This stablity metric was computed over 10.000 samples from the model, for a detailed explanation of stability see Appendix A.3. Observe that it might also be possible to utilize post-processing to increase the molecule stability using prior knowledge. However, here we are mainly using this metric to see how many stable atoms and molecules the E-NF is able to generate in one shot, only by learning the molecule distribution. The E-NF on average produces 85% valid atoms, whereas the best baseline only produces 75% valid atoms. An even stronger improvement is visible when comparing molecule stability: where the E-NF produces 4.9% stable molecules versus 0.5% by the best baseline. Interestingly, the percentage of stable molecules is much lower than that of atoms. This is not unexpected: if even one atom in a large molecule is unstable, the entire molecule is considered to be unstable.

Finally, we evaluate the Validity, Uniqueness and Novelty as defined in (Simonovsky and Komodakis, 2018) for the generated molecules that are stable. For this purpose, we map the 3-dimensional representation of stable molecules to a graph structure and then to a SMILES notation using the rdkit toolkit. All our stable molecules are already defined as valid, therefore we only report the Novelty and Uniqueness since the Validity of those molecules that are already stable is 100%. The Novelty is defined as the ratio of stable generated molecules not present in the training set and Uniqueness is the ratio of unique stable generated molecules. Notice that different generated point clouds in 3-dimensional space may lead to the same molecule in the graph space or SMILES notation. Therefore, even if in the 3D space, all our generated samples were unique and novel, the underlying molecule that they represent doesn't have to be. Using our E-NF, we generated 10.000 examples to compute these metrics. We obtained 491 stable molecules (4.91 %), from these subset 415 molecules were novel (84.4%) and 352 were unique (71.8%). Further details and analyses are provided in Appendix A.4.

**Results (qualitative):** In Figure 5 samples from our model are shown. The top row contains random samples that have not been cherry-picked in any way. Note that although the molecule structure is mostly accurate, sometimes small mistakes are visible: Some molecules are disconnected, and some atoms in the molecules do not have the required number of bonds. In the bottom row, random samples have been visualized but under the condition that the sample is stable. Note that atoms might be double-bonded or triple-bonded, which is indicated in the visualization software. For example, in the molecule located in the bottom row 4th column, an oxygen atom is double bonded with the carbon atom at the top of the molecule.

## 6 Limitations and Conclusions

**Limitations**  In spite of the good results there are some limitations in our model that are worth mentioning and could be addressed in future work: 1) The ODE type of flow makes the training computationally expensive since the same forward operation has to be done multiple times sequentially in order to solve the ODE equation. 2) The combination of the ODE with the EGNN exhibited some instabilities that we had to address (Equation 9). Despite the model being stable in most of the experiments, we still noticed some rare peaks in the loss of the third experiment (QM9) that in one very rare case made it diverge. 3) Different datasets may also contain edge data for which E-NFs should be extended. 4) Our likelihood estimation is invariant to reflections, therefore our model assigns the same likelihood to both a molecule and its mirrored version, which for chiral molecules may not be the same.

**Societal Impact**  Since this work presents a generative model that can be applied to molecular data, it may advance the research at the intersection of chemistry and deep learning. In the long term, more advanced molecule generation methods can benefit applications in drug research and material discovery. Those long term applications may have a positive impact in society, for example, creating new medications or designing new catalyst materials that permit cheaper production of green energy. Negative implications of our work may be possible when molecules or materials are created for immoral purposes, such as the creation of toxins or illegal drugs.

**Conclusions**  In this paper we have introduced E($n$) Equivariant Normalizing Flows (E-NFs): A generative model equivariant to Euclidean symmetries. E-NFs are continous-time normalizing flows that utilize an EGNN with improved stability as parametrization. We have demonstrated that E-NFs considerably outperform existing normalizing flows in log-likelihood on DW4, LJ13, QM9 and also in the stability of generated molecules and atoms.

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
