# A Experiment details

In this Appendix section we provide more details about the imlementation of the experiments. First we introduce those parts of the model architecture that are the same across all experiments. The EGNN module defined in Section 2.1, Equations 4 and 5 is composed of four Multilayer Perceptrons (MLPs) $\phi_e$, $\phi_x$, $\phi_h$ and $\phi_{\text{inf}}$. We used the same design from the original EGNN paper (Satorras et al., 2021) for all MLPs except for $\phi_x$. The description of the modules would be as follows:

- $\phi_e$ (edge operation): consists of a two-layer MLP with two SiLU (Nwankpa et al., 2018) activation functions that takes as input $(\mathbf{h}_i, \mathbf{h}_j, \|\mathbf{x}_i - \mathbf{x}_j\|^2)$ and outputs the edge embedding $\mathbf{m}_{ij}$.

- $\phi_x$ (coordinate operation): consists of a two layers MLP with a SiLU activation function in its hidden layer and a Tanh activation function at the output layer. It takes as input the edge embedding $\mathbf{m}_{ij}$ and outputs a scalar value.

- $\phi_h$ (node operation): consists of a two layers MLP with one SiLU activation function in its hidden layer and a residual connection: $[\mathbf{h}_i^l, \mathbf{m}_i] \to \{\text{Linear}() \to \text{SiLU}() \to \text{Linear}() \to \text{Addition}(\mathbf{h}_i^l)\} \to \mathbf{h}_i^{l+1}$.

- $\phi_{\text{inf}}$ (edge inference operation): Connsists of a Linear layer followed by a Sigmoid layer that takes as input the edge embedding $\mathbf{m}_{ij}$ and outputs a scalar value.

These functions define our E(n) Equivariant Flow dynamics (E-NF), the Graph Normalizing Flow dynamics (GNF), Graph Normalizing Flow with attention (GNF-att) and Graph Normalizing Flow with attention and data augmentation (GNF-att-aug). The variations regarding architectural choices among experiments are the number of layers and hidden features per layer. In the following we explicitly write down the dynamics used for the ENF, GNF, GNF-att and GNF-att-aug baselines.

- E-NF (Dynamics): We can write the ENF dynamics introduced in Section 2.1 adapted for our method in Section 4 as:

$$\mathbf{m}_{ij} = \phi_e\left(\mathbf{h}_i^l, \mathbf{h}_j^l, \left\|\mathbf{x}_i^l - \mathbf{x}_j^l\right\|^2\right) \tag{13}$$

$$\mathbf{x}_i^{l+1} = \mathbf{x}_i^l + \sum_{j \neq i} \frac{(\mathbf{x}_i^l - \mathbf{x}_j^l)}{\|\mathbf{x}_i^l - \mathbf{x}_j^l\| + 1} \phi_x\left(\mathbf{m}_{ij}\right) \tag{14}$$

$$\mathbf{m}_i = \sum_{j \neq i} \phi_{\text{inf}}(\mathbf{m}_{ij})\mathbf{m}_{ij}, \tag{15}$$

$$\mathbf{h}_i^{l+1} = \phi_h\left(\mathbf{h}_i^l, \mathbf{m}_i\right) \tag{16}$$

- GNF: The dynamics for this method are a standard Graph Neural Network which can also be interpreted as a variant of the EGNN with no equivariance. The dataset coordinates $\mathbf{x}$ are treated as $\mathbf{h}$ features, therefore they are provided as input to $\mathbf{h}^0$ through a linear mapping before its first layer. Since our datasets do not consider adjacency matrices we let $e_{ij} = 1$ for all $ij$.

$$\mathbf{m}_{ij} = \phi_e\left(\mathbf{h}_i^l, \mathbf{h}_j^l\right) \tag{17}$$

$$\mathbf{m}_i = \sum_{j \neq i} e_{ij}\mathbf{m}_{ij}, \tag{18}$$

$$\mathbf{h}_i^{l+1} = \phi_h\left(\mathbf{h}_i^l, \mathbf{m}_i\right) \tag{19}$$

- GNF-att: This model is almost the same as GNF with the only difference that it infers the edges through $e_{ij} = \phi_{\text{inf}}(\mathbf{m}_{ij})$ which can also be seen as a form of attention.

- GNF-att-aug: This is the exact same model as GNF-att. The only difference lies in the pre-processing of the data since we perform data augmentation by rotating the node positions before inputting them to the model.

## A.1 DW4 and LJ13 experiments

In the following we report the DW4 (equation 20) and LJ13 (equation 21) energy functions introduced in (Köhler et al., 2020):

$$u^{\text{DW}}(x) = \frac{1}{2\tau} \sum_{i,j} a \left( d_{ij} - d_0 \right) + b \left( d_{ij} - d_0 \right)^2 + c \left( d_{ij} - d_0 \right)^4 \tag{20}$$

$$u^{\text{LJ}}(x) = \frac{\epsilon}{2\tau} \left[ \sum_{i,j} \left( \left( \frac{r_m}{d_{ij}} \right)^{12} - 2 \left( \frac{r_m}{d_{ij}} \right)^6 \right) \right] \tag{21}$$

Where $d_{ij} = \|\mathbf{x}_i - \mathbf{x}_j\|$ is the distance between two particles. The design parameters $a$, $b$, $c$, $d$ and temperature $\tau$ from DW4 and the design parameters $\epsilon$, $r_m$ and $\tau$ from LJ13 are the same ones used in (Köhler et al., 2020).

**Implementation details** All methods are trained with the Adam optimizer, weight decay $10^{-12}$, batch size 100, the learning rate was tuned independently for each method which resulted in $10^{-3}$ for all methods except for the E-NF model which was $5 \cdot 10^{-4}$.

In tables 3 and 4 we report the same DW4 and LJ13 averaged results from Section 5.1 but including the standard deviations over the three runs.

Table 3: Negative Log Likelihood comparison on the test partition of DW4 dataset for different amount of training samples. Averaged over 3 runs and including standard deviations.

| # Samples | DW4 | | | |
| --- | --- | --- | --- | --- |
| | $10^2$ | $10^3$ | $10^4$ | $10^5$ |
| GNF | $11.93 \pm 0.41$ | $11.31 \pm 0.07$ | $10.38 \pm 0.11$ | $7.95 \pm 0.17$ |
| GNF-att | $11.65 \pm 0.39$ | $11.13 \pm 0.38$ | $9.34 \pm 0.29$ | $7.83 \pm 0.15$ |
| GNF-att-aug | $8.81 \pm 0.23$ | $8.31 \pm 0.19$ | $7.90 \pm 0.04$ | $7.61 \pm 0.06$ |
| Simple dynamics | $9.58 \pm 0.05$ | $9.51 \pm 0.01$ | $9.53 \pm 0.02$ | $9.47 \pm 0.06$ |
| Kernel dynamics | $8.74 \pm 0.02$ | $8.67 \pm 0.01$ | $8.42 \pm 0.00$ | $8.26 \pm 0.03$ |
| E-NF | $\mathbf{8.31} \pm 0.05$ | $\mathbf{8.15} \pm 0.10$ | $\mathbf{7.69} \pm 0.06$ | $\mathbf{7.48} \pm 0.05$ |

Table 4: Negative Log Likelihood comparison on the test partition of LJ13 dataset for different amount of training samples. Averaged over 3 runs and including standard deviations.

| # Samples | LJ13 | | | |
| --- | --- | --- | --- | --- |
| | 10 | $10^2$ | $10^3$ | $10^4$ |
| GNF | $43.56 \pm 0.79$ | $42.84 \pm 0.52$ | $37.17 \pm 1.79$ | $36.49 \pm 0.81$ |
| GNF-att | $43.32 \pm 0.20$ | $36.22 \pm 0.34$ | $33.84 \pm 1.60$ | $32.65 \pm 0.57$ |
| GNF-att-aug | $41.09 \pm 0.53$ | $31.50 \pm 0.35$ | $30.74 \pm 0.86$ | $30.93 \pm 0.73$ |
| Simple dynamics | $33.67 \pm 0.07$ | $33.10 \pm 0.10$ | $32.79 \pm 0.13$ | $32.99 \pm 0.11$ |
| Kernel dynamics | $35.03 \pm 0.48$ | $31.49 \pm 0.06$ | $31.17 \pm 0.05$ | $31.25 \pm 0.12$ |
| E-NF | $\mathbf{33.12} \pm 0.85$ | $\mathbf{30.99} \pm 0.95$ | $\mathbf{30.56} \pm 0.35$ | $\mathbf{30.41} \pm 0.16$ |

## A.2 QM9 Positional and QM9

Both experiments QM9 and QM9 positional have been trained with batch size 128 and weight decay $10^{-12}$. The learning rate was set to $5 \cdot 10^{-4}$ for all methods except for the E-NF and Simple dynamics where it was reduced to $2 \cdot 10^{-12}$.

The flows trained on QM9 have all been trained for 30 epochs. Training these models takes approximately 2 weeks using two NVIDIA 1080Ti GPUs. The flows trained on QM9 Positional have been trained for 160 epochs in single NVIDIA 1080Ti GPUs. Simple Dynamics would train in less than a day, Kernel Dynamics around 2 days, the other methods can take up to 7 days. The training of the models becomes slower per epoch as the performance improves, due to the required steps in the ODE solver. For QM9, the model performance is averaged over 3 test set passes, where variance originates from dequantization and the trace estimator, see Table 5.

Table 5: Neg. log-likelihood averaged over 3 passes, variance from dequantization and trace estimator.

|  | NLL |
| --- | --- |
| **GNF-attention** | -28.2 $\pm$ 0.49 |
| **GNF-attention-augmentation** | -29.3 $\pm$ 0.02 |
| **E-NF** (ours) | -59.7 $\pm$ 0.12 |

### A.3   Stability of Molecules Benchmark

In the QM9 experiment, we also report the % of stable molecules and atoms. This section explains how we test for molecule stability. In addition, we explain why there is not a set of rules that will judge *every* molecule in the dataset stable. First of all, we say that an atom is stable when the number of bonds with other atoms matches their valence. For the atoms used their respective valencies are (H: 1, C: 4, N: 3, O: 2, F: 1). A molecule is stable when *all* of its atoms are stable. The most straightforward method to decide whether atoms are bonded is to compare their relative distance. There are some limitations to this method. For instance, QM9 contains snapshots of molecules in a single configuration, and in reality atoms of a molecule are constantly in motion. In addition, the type of molecule can also greatly influence the relative distances, for instance due to collisions, the Van der Waals force and inter-molecule Hydrogen bonds. Further, environmental circumstances such as pressure and temperature may also affect bond distance. For these reasons it is not possible to design a distance based rule that considers every molecule in QM9 stable, based only on a snapshot. To find the most optimal rules for QM9, we tune the average bond distance for every atom-type pair to achieve the highest molecule stability on QM9 on the train set with results in 95.3% stable molecules and 99% stable atoms. On the test set these rules result in 95.2% stable molecules and 99% stable atoms.

The specific distances that we used to define the types of bond (SINGLE, DOUBLE TRIPLE or NONE) are available in the code and were obtained from `http://www.wiredchemist.com/chemistry/data/bond_energies_lengths.html`. Notice the type of bond depends on the type of atoms that form that bond and the relative distance among them. Therefore, given a conformation of atoms, we deterministically compute the bonds among all pairs of atoms. Then we say an atom is stable if its number of bonds with other atoms matches its valence.

### A.4   Further QM9 analysis | Validity, Uniqueness, Novelty

*Validity* is defined as the ratio of molecules that are valid from all the generated ones. *Uniqueness* is defined as the number of valid generated molecules that are unique divided by the number of all valid generated molecules. *Novelty* is defined as the number of valid generated molecules that are not part of the training set divided by the total number of valid generated molecules. In our case, all stable molecules are valid, therefore, we only report the Uniqueness and Novelty. Absolute and percentage values are reported when generating 10.000 examples. Metrics have been computed as in (Simonovsky and Komodakis, 2018) with rdkit `https://www.rdkit.org/`.

In addition to the previous metrics, in Figure 6, we plot a histogram of the number of atoms per molecule and also of the type of atoms for both the stable generated molecules and the ground truth ones.

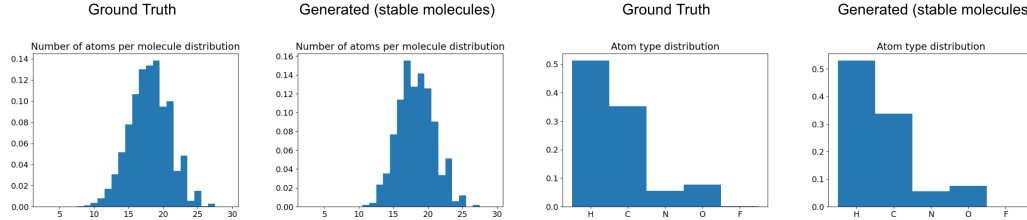

Figure 6: Number of atoms per molecule distributions and atom type distribution for the stable generated molecules and the training (ground truth) molecules.

# B  Lifting Discrete Features to Continuous Space

In this section additional details are discussed describing how discrete variables are lifted to a continuous space. As in the main text, we let $\mathbf{h} = (\mathbf{h}_{\text{ord}}, \mathbf{h}_{\text{cat}})$ be the discrete features on the nodes, either ordinal or categorical. For simplicity we can omit the number of nodes and even the number of feature dimensions. Then $\mathbf{h}_{\text{ord}} \in \mathbb{Z}$ and $\mathbf{h}_{\text{cat}} \in \mathbb{Z}$. Note here that although the representation for $\mathbf{h}_{\text{ord}}$ and $\mathbf{h}_{\text{cat}}$ are the same, they are treated differently because of their ordinal or categorical nature.

Now let $\boldsymbol{h} = (\boldsymbol{h}_{\text{ord}}, \boldsymbol{h}_{\text{cat}})$ be its continuous counterpart. We will utilize variational dequantization (Ho et al., 2019) for the ordinal features. In this framework mapping $\boldsymbol{h}_{\text{ord}}$ to $\mathbf{h}_{\text{ord}}$ can be done via rounding (down) so that $\mathbf{h}_{\text{ord}} = \text{round}(\boldsymbol{h}_{\text{ord}})$. Similarly we will utilize argmax flows (Hoogeboom et al., 2021) to map the categorical map which amounts to $\mathbf{h}_{\text{cat}} = \text{argmax}(\boldsymbol{h}_{\text{cat}})$. Here $\boldsymbol{h}_{\text{ord}} \in \mathbb{R}$ and $\boldsymbol{h}_{\text{cat}} \in \mathbb{R}^K$ where $K$ is the number of classes.

The transformation $\boldsymbol{h} \mapsto \mathbf{h}$ given by $\mathbf{h} = (\text{round}(\boldsymbol{h}_{\text{ord}}), \text{argmax}(\boldsymbol{h}_{\text{cat}}))$ is completely deterministic, and to derive our objective later we can formalize this as a distribution with all probability mass on a single event: $P(\mathbf{h}|\boldsymbol{h}) = \mathbb{1}[\mathbf{h} = (\text{round}(\boldsymbol{h}_{\text{ord}}), \text{argmax}(\boldsymbol{h}_{\text{cat}}))]$ as done in (Nielsen et al., 2020). Then the (discrete) generative model $p_{\text{H}}(\mathbf{h}) = \mathbb{E}_{\boldsymbol{h} \sim p_H(\boldsymbol{h})} P(\mathbf{h}|\boldsymbol{h})$ defined via the continuous $p_H(\boldsymbol{h})$ can be optimized using variational inference:

$$\log p_{\text{H}}(\mathbf{h}) \geq \mathbb{E}_{\boldsymbol{h} \sim q_{\text{ord,cat}}(\,\cdot\,|\mathbf{h})} \left[ \log p_H(\boldsymbol{h}) - \log q_{\text{ord,cat}}(\boldsymbol{h}|\mathbf{h}) + \log P(\mathbf{h}|\boldsymbol{h}) \right]$$

$$= \mathbb{E}_{\boldsymbol{h} \sim q_{\text{ord,cat}}(\,\cdot\,|\mathbf{h})} \left[ \log p_H(\boldsymbol{h}) - \log q_{\text{ord,cat}}(\boldsymbol{h}|\mathbf{h}) \right],$$

for which we need a distribution $q_{\text{ord,cat}}(\boldsymbol{h}|\mathbf{h})$ that has support only where $P(\mathbf{h}|\boldsymbol{h}) = 1$. In other words, $q_{\text{ord,cat}}$ needs to be the probabilistic inverse of $P(\mathbf{h}|\boldsymbol{h})$, so the round and arg max functions.

So how do we ensure that $q_{\text{ord,cat}}$ is the probabilistic inverses of the round and argmax functions? For the ordinal data we construct a distribution as follows. First the variable $\boldsymbol{u}$ is distributed as a Gaussian, $\boldsymbol{u}_{\text{logit}} \sim \mathcal{N}(\,\cdot\,|\mu(\mathbf{h}_{\text{ord}}), \sigma(\mathbf{h}_{\text{ord}}))$ where the mean and standard deviation are predicted by a shared EGNN (denoted by $\mu$ and $\sigma$) with the discrete features as input. Then $\boldsymbol{u} = \text{sigmoid}(\boldsymbol{u}_{\text{logit}})$ ensuring that $\boldsymbol{u} \in (0,1)$. We will name this distribution $q(\boldsymbol{u}|\mathbf{h}_{\text{ord}})$. This construction is practical because we can compute $\log q_{\text{ord}}(\boldsymbol{u}|\mathbf{h}_{\text{ord}}) = \log \mathcal{N}(\boldsymbol{u}_{\text{logit}}|\mu(\mathbf{h}_{\text{ord}}), \sigma(\mathbf{h}_{\text{ord}})) - \log \text{sigmoid}'(\boldsymbol{u}_{\text{logit}})$ using the change of variables formula. Finally we let $\boldsymbol{h}_{\text{ord}} = \mathbf{h}_{\text{ord}} + \boldsymbol{u}$, for which we write the corresponding distribution as $q_{\text{ord}}(\boldsymbol{h}_{\text{ord}}|\mathbf{h}_{\text{ord}})$. This last step only shifts the distribution and so does not result in a volume change, so $\log q_{\text{ord}}(\boldsymbol{h}_{\text{ord}}|\mathbf{h}_{\text{ord}}) = \log q(\boldsymbol{u}|\mathbf{h}_{\text{ord}})$. Now since $\boldsymbol{u} \in (0,1)$ we have that $\text{round}(\boldsymbol{h}_{\text{ord}}) = \text{round}(\mathbf{h}_{\text{ord}} + \boldsymbol{u}) = \mathbf{h}$ as desired.

For categorical data we similarly model a unconstrained noise variable $\boldsymbol{w} \sim \mathcal{N}(\,\cdot\,|\mu(\mathbf{h}_{\text{cat}}), \sigma(\mathbf{h}_{\text{ord}}))$ which is then transformed to respect the argmax contraint. Again, $\mu$ and $\sigma$ are modelled by an EGNN. Let $i = \mathbf{h}_{\text{cat}}$, the index whos value needs to be the maximum and $\boldsymbol{w}_i = T$. Then $\boldsymbol{h}_{\text{cat},i} = T$ and $\boldsymbol{h}_{\text{cat},-i} = T - \text{softplus}(T - \boldsymbol{w}_{-i})$. Again the log-likelihood of the corresponding of the resulting distribution $q_{\text{cat}}(\boldsymbol{h}_{\text{cat}}|\mathbf{h}_{\text{cat}})$ is computed using the $\log \mathcal{N}(\boldsymbol{w}|\mu(\mathbf{h}_{\text{cat}}, \sigma(\mathbf{h}_{\text{ord}})))$ and the log derivatives of the softplus thresholding. For more details on these constructions see (Ho et al., 2019; Hoogeboom et al., 2021).