# OpenReview forum: "E(n) Equivariant Normalizing Flows"
_NeurIPS.cc/2021/Conference — NeurIPS 2021 Oral_

### Official Review · Reviewer_iDHj · 2021-07-16

**Rating:** 7
**Confidence:** 4

**Summary:**

The authors introduce a novel approach for generating data with intrinsic Euclidean symmetries. The proposed model uses recent advances in continuous normalizing flows, variational dequantization, and equivariant neural networks to construct a generative algorithm E(n)-equivariant along some modeled variables. The authors tackled many theoretical and technical problems to make the model stable and obtain desired equivariance in Euclidean space. The generative process looks as follows: first, the two groups of latent codes are sampled from the Gaussian distribution, which is rotation and reflection invariant; these samples are passed through the continuous normalizing flow guided by an Equivariant Graph Neural Networks. To force the translation equivariance, the authors propose to center the latent codes and work with (M-1) x n linear subspace. The target distribution is given by applying the variational dequantization for the sampled latent codes. The authors validate their approach on synthetic data and molecular conformations produced with DFT (density functional theory).

**Limitations And Societal Impact:**

The limitation of the work lies in the simplicity of the provided experiments on molecular data. First, important to say that model tries to solve two distinct drug discovery problems simultaneously — learn the distribution of molecular compounds [1] and model the distribution of conformational space for the molecular compounds [2]. Both of these problems have a wide variety of developed methods (including the neural-based) and a set of metrics on which the methods compete. I would recommend comparing the proposed model with the state-of-the-art model from these areas on the set of distribution learning and conformation modeling metrics.

[1] Molecular Sets (MOSES): A Benchmarking Platform for Molecular Generation Models, Polykovskiy et al.
[2] Learning Neural Generative Dynamics for Molecular Conformation Generation, Xu et al.


**Main Review:**

**Originality:** Although all key parts are well-known, combining everything in one framework seems novel and original.

**Quality:** The authors motivate this work well, describe all important details of their method, and propperly discuss limitations and impact of their work. The experiments on molecular data are quite simple and miss several important metrics to evaluate whether the proposed method can compete with state-of-the-art approaches in drug discovery.

**Clarity:** The paper is well written and is easy to follow. The authors cover all key details of the proposed method and discuss all problems that were solved while developing the method.

**Significance:** The paper provides simple experiments that do not evaluate the practical significance of the method in real drug discovery problems. The missing standard deviation in metrics also makes it difficult to validate significance of the results.

Update: The authors clarified my concerns, I tend to keep the original score

**Time Spent Reviewing:**

6

---

> ### Author Response · Authors · 2021-08-09
> **Response to Reviewer iDHj**
>
> First of all, thank you for the detailed comments and positive feedback indicating that we present a novel and original framework in a well written manner. To address the remarks:
>
> >The experiments on molecular data are quite simple and miss several important metrics to evaluate whether the proposed method can compete with state-of-the-art approaches in drug discovery.
>
> To expand our analysis for the molecule generation experiment, we are planning to include a uniqueness and novelty experiment, as suggested by reviewer aifD. Qualitatively, we already see that there is a wide variety between the samples taken from the model. With these additional experiments we aim to report the uniqueness and the novelty with respect to the training set of the generated molecules. Additionally, we will report the first-order statistics for nodes and bonds of the generated samples which will give another indication of how well the distribution is learned, when compared to these statistics from the data.
>
> >The missing standard deviation in metrics also makes it difficult to validate significance of the results.
>
> The standard deviations for experiments 1 and 2 are reported in Appendix A.1. The last experiment takes much longer to train such that we could only obtain the results of a single run per model.

---

### Official Review · Reviewer_aifD · 2021-07-16

**Rating:** 7
**Confidence:** 4

**Summary:**

The authors introduce a equivariant model for one-shot 3D molecule generation, building upon continuous time normalizing flows and equivariant graph networks. They demonstrated the utility of their approach based on several benchmark applications where they consistently outperform other flow-based models.

**Limitations And Societal Impact:**

Potential limitations and societal impact have been sufficiently addressed.


**Main Review:**

The proposed model present a sensible combination of existing methods, namely E(n) equivariant graph neural networks, variational dequantization and argmax flows. This enables the model to lean distributions over molecules considering positions as well as atom types. This is a clear improvement over previous related normalizing flows, which were focused on positional data.

The reported molecular generation experiments serve as a clear demonstration for the benefits of their method over other normalizing flows. However, one criticism is the strong focus on this particular class of methods  which is particularly noticeable in the final QM9 experiment. This makes it hard to evaluate how the proposed model behaves with respect to other generative approaches for 3D molecular structure, many of which have even been applied to the same dataset. Examples include iterative approaches that place atom after atom [1,2,3] as well as other one-shot approaches, either GAN- [4] or VAE-based [5]. A comparison to other methods is also made difficult by using the negative log-likelihood as primary evaluation metric. In this context, it would be worthwhile to analyze other statistics of the structures sampled in the QM9 experiment, e.g. how many of the stable molecules (which pass a valency check of choice) are training structures, test structures, or even novel structures and how many are unique? Additional measures of interest would be how the ring/atom/bond count of the generated structures compare to their distribution in QM9. One additional question would be how suitable the model is for practical applications, as the number of valid sampled molecules appear low at a first glance? Since QM9 provides relaxed structures of the molecules, one measure to consider in future experiments is the median RMSD between atomic positions of generated and relaxed structures, i.e.  how close are the sampled structures to actual equilibrium configurations (a drawback of this metric is that it is computationally intensive due to the need for quantum chemical computations).

The paper itself is well written providing a thorough description of the formal/theoretical background as well as the architecture.

On the whole, the proposed approach build upon and extends an interesting combination of machine learning approaches and represents a viable contribution with respect to one-shot generation. What remains unclear how it performs in comparison to other, non flow, generative approaches.

1 Simm, Gregor, Robert Pinsler, and José Miguel Hernández-Lobato. "Reinforcement learning for molecular design guided by quantum mechanics." International Conference on Machine Learning. PMLR, 2020.

2 Simm, Gregor NC, et al. "Symmetry-aware actor-critic for 3d molecular design." arXiv preprint arXiv:2011.12747 (2020).

3 Gebauer, N. W., Gastegger, M., & Schütt, K. T. (2019). Symmetry-adapted generation of 3d point sets for the targeted discovery of molecules. arXiv preprint arXiv:1906.00957.

4 Hoffmann, Moritz, and Frank Noé. "Generating valid Euclidean distance matrices." arXiv preprint arXiv:1910.03131 (2019).

5 Nesterov, Vitali, Mario Wieser, and Volker Roth. "3DMolNet: a generative network for molecular structures." arXiv preprint arXiv:2010.06477 (2020).


**Time Spent Reviewing:**

3

---

> ### Author Response · Authors · 2021-08-09
> **Response to Reviewer aifD**
>
> First of all, thank you for the detailed comments and positive feedback indicating that we present a clear improvement over previous related normalizing flows. We answer the raised issues below.
>
> The reviewer mentions that a comparison to other methods is made difficult by using the negative log-likelihood as a main metric, and it would be worthwhile to include other metrics as an analysis of the uniqueness and novelty of generated molecules.
> Qualitatively we see that the sampled molecules are very diverse. This is expected, because the flow is optimized on a negative log-likelihood, which is equivalent to a forward KL minimization. Because such objectives do not collapse to modes, this ensures diversity in samples. Nonetheless, we agree that an additional analysis on uniqueness and novelty would be helpful, for instance by converting the samples to a schematic representation and then computing the uniqueness and novelty in that space. We will include this in an updated version of the paper.
>
> The reviewer also asks about first order statistics of the generated samples, such as bond or atom distributions. Note that we reported the relative distances distribution in the “QM9 Positional” dataset, and we will extend this measure to the QM9 experiment for the bond and atom distributions.
>
> Finally, the reviewer proposes to compute the RMSD between the atomic positions of generated structures and actual equilibrium configurations, but he mentions this metric can be computationally intensive due to the need of quantum chemical computations. Agreeing with the reviewer, we think this metric may be too hard to compute since finding the closest equilibrium configuration to a generated structure is still an open and very complex problem. Additionally, RMSD and similar metrics promote mode-seeking behaviour, whereas the aim of our method is to learn distributions over positions that may fluctuate.

---

> > ### Comment · Reviewer_aifD · 2021-08-20
> > **Response to rebuttal**
> >
> > Thank you for the discussion and inclusion of additional analysis. Some of my concerns regarding analysis/practical applicability remain, and I hope future evaluations of the method will also include validation with electronic structure theory.
> >
> > In this context, I would like to make a few comments. In a QM9 setting, the closest equilibrium structure can be found with a standard structure optimization, since one is typically only interested in the nearest local minimum. While this is to costly for the rebuttal phase, it can be done in a reasonable amount of time by pre-optimising with a cheaper electronic structure method (e.g. GGA functional). I also agree that in general computing the RMSD with respect to the closest minimum might not be a suitable analysis, but in the case of QM9 it is assumed that all structures are equilibrium structures. In a general setting, energies will be a more reliable measure. QM9 was computed with the Gaussian version of the B3LYP functional using the 6-31G(2df,2p) Pople basis set, which means that there is a well defined standard for evaluations. If one is interested in the stability of the molecules, the simplest method would be to compute the B3LYP energies of the generated structures and compare to the the sum of the free atom energies (which corresponds to the dissociation limit). A more reliable way is to relax to the nearest minimum and e.g. compare the energy difference between relaxed and original structure to the dissociation limit. Bond and graph based analysis should always be taken with a grain of salt, as they rely on empirical definitions of bonds and bond orders.
> >
> > Despite these concerns, I believe the proposed method has potential and my rating remains at 7.

---

### Official Review · Reviewer_fcXk · 2021-07-28

**Rating:** 7
**Confidence:** 3

**Summary:**

The paper introduces a new type of equivariant normalizing flow by using the CNF framework with a modified equivariant graph NN as a dynamics function. The equivariance discussed is wrt Euclidean symmetries, which are important for physical systems. They show that their flow beats prior equivariant baselines on synthetic data. Moreover, the model is capable of transforming spacial, categorical and ordinal features simultaneously. This allows the sampling of molecular configurations with positions, atom types, and charges.

**Limitations And Societal Impact:**

As mentioned above, a further limitation could be that the generated molecules are unstable considering their energies. So far, this is not mentioned in the paper.
The authors adequately addressed potential negative societal impact of their work.


**Main Review:**

Merits of paper
- The paper is clearly written, well organized and easy to understand.
- They improve on previous work of Köhler et al. (2019, 2020) by utilizing equivariant graph NN as a dynamic function instead of their "Kernel Dynamics" function.
- Incorporating equivariant graph NN into the (equivariant) continuous normalizing flow framework is non-trivial and novel.
- The theory and related work are well presented. All previous work is acknowledged when necessary.
- They compare to prior baseline of Köhler et al. (2020) and show that their model performs superior.
- They introduce a new benchmark dataset, based on the QM9 dataset, and show that their model outperforms other non-equivariant as well as equivariant models.
- The theory of equivariant flows is heavily based on the derivation from Köhler et al. (2019, 2020) and Rezende et al. (2019), but they contribute a proof for equivalence of the Jacobian of the used subspace and the whole space.
- They are able to sample new molecular configurations of different size with corresponding atom types and charges simultaneously. Meaning, they were (at least to some extent) able to learn the distribution of the subset of QM9 molecules with their E-NF.
- This is a valuable contribution, sets a new baseline for molecular structure generation, and is very interesting for researchers in that area.

Criticism
- Although the training process and network architecture is outlined quite well, I could not find information about the Jacobian evaluation. This is of interest as it has proven to be a bottleneck in prior work, e.g. Köhler et al. (2020).  I would assume that some sort of approximation, such as the Hutchinson estimator, is used at least in the training process. I think this information should be included in the paper (or at least in the appendix).
- The most interesting experiment is the generation of molecules. However, their stability measure is only based on distances. I think a better metric for stability measure would be the energy of the generated molecules. In fact, it might be that far less than 5% of the generated molecules are stable considering the energy instead of the distances, as already small deviations of the distances as well as the atom types can have huge impact on the energies. This should at least be discussed in the final paper.

All in all, a good paper, especially when the energy of the generated molecules is included.

**Time Spent Reviewing:**

8

---

> ### Author Response · Authors · 2021-08-09
> **Response to reviewer fcXk**
>
> First of all, thank you  for the detailed comments and positive feedback indicating that the presented framework is non-trivial, novel and that the paper is clearly written and easy to understand. We answer the raised issues below.
>
> > I could not find information about the Jacobian evaluation. I would assume that some sort of approximation, such as the Hutchinson estimator, is used at least in the training process.
>
> Indeed, following existing approaches in image modelling, we use Hutchinson’s estimator to estimate the trace of the Jacobian in both training and test settings. We mentioned the estimator in the background section but as pointed out by the reviewer we did not specify precisely that we are using it in our method. We will include it in the main description of the model (Section 4).
>
> > The most interesting experiment is the generation of molecules. However, their stability measure is only based on distances. I think a better metric for stability measure would be the energy of the generated molecules. In fact, it might be that far less than 5% of the generated molecules are stable considering the energy instead of the distances, as already small deviations of the distances as well as the atom types can have huge impact on the energies. This should at least be discussed in the final paper.
>
> We agree the energy of generated molecules would be an ideal metric, but there is not a standard method to compute it for QM9 molecules. If the reviewer has a specific method in mind, then please feel free to let us know in the discussion phase. In addition, we would like to point out that our stability metric is not only considering fixed distances, but is in fact already considering the type of bonds between atoms (single, double or triple bonds) which is directly correlated with the energy levels of the molecule. Our metric is already very sensitive to small distance changes, and molecules are only considered valid if the distances between specific atom types are in very small predefined intervals.
>
> We think this point raised by the reviewer is very relevant, and even if the stability metric is included in the code, we did not explain it with enough detail in the Appendix. We will include an extensive explanation of this metric, and also a table from chemistry of all the accounted bond types given the distances for each pair of atom types.

---

> > ### Comment · Reviewer_fcXk · 2021-09-13
> > **Response to rebuttal**
> >
> > Thank you for including the additional information about the Hutchinson’s estimator and especially the stability measure in the final paper.
> > I agree that it is difficult to find a suitable evaluation metric, and I am looking forward to the detailed explanation of your method in the final paper.
> >
> > My rating remains the same.

---

### Decision · Program_Chairs · 2021-09-27

**Decision:**

Accept (Oral)

**Comment:**

All reviewers agree on acceptance. This work is a highly significant contribution by introducing equivariant graph neural networks into the
(equivariant) continuous normalizing flow framework to obtain invertible equivariant functions. They show that their flow beats prior equivariant baselines and allows the sampling of molecular configurations with positions, atom types, and charges. A clear acceptance decision.

Pros:

- High significant contribution using equivariant graph neural networks and continuous normalizing flows to obtain invertible equivariant functions.
- The model can lean distributions over molecules considering positions as well as atom types, a clear improvement over previous related normalizing flows, which were focused on positional data.
- The reported molecular generation experiments serve as a clear demonstration for the benefits of their method over other normalizing flows.
- The theory of equivariant flows is heavily based on the derivation from Köhler et al. (2019, 2020) and Rezende et al. (2019), but they contribute a proof for equivalence of the Jacobian of the used subspace and the whole space.
- Sets a new baseline for molecular structure generation.
- Very well-written paper. The authors motivate this work well, describe all important details of their method, and properly discuss limitations and impact of their work.

Limitations:

- Remains unclear how it performs in comparison to other, nonflow, generative approaches.
- Energies of the generated molecules are not reported.

The above list of pros and few existing limitations make this work highly relevant and worth an oral presentation.